miRMOD: a tool for identification and analysis of 5′ and 3′ miRNA modifications in Next Generation Sequencing small RNA data

Kaushik Abhinav 1
Saraf Shradha 1
Mukherjee Sunil K. 2
Gupta Dinesh 1 dinesh@icgeb.res.in
1 Bioinformatics Laboratory, SCB Group, International Centre for Genetic Engineering & Biotechnology , New Delhi , India
2 Department of Genetics, University of Delhi, South Campus , Delhi , India
Papaleo Elena
Electronic publication date: 2015 Oct 20
Publication date: 2015
Volume: 3
Electronic Location ID: e1332
Received 2015 Jul 30; Accepted 2015 Sep 27
Copyright: © 2015 Kaushik et al.
Copyright year: 2015
Copyright holder: Kaushik et al.
License: This is an open access article distributed under the terms of the Creative Commons Attribution License, which permits unrestricted use, distribution, reproduction and adaptation in any medium and for any purpose provided that it is properly attributed. For attribution, the original author(s), title, publication source (PeerJ) and either DOI or URL of the article must be cited.
License URL: https://creativecommons.org/licenses/by/4.0/

Keywords: 3′ and 5′ modifications, miRNA, Next Generation Sequencing (NGS), Target alteration, Non-templated additions, Trimming

Funding: Department of Biotechnology (DBT, India) ICGEB BT/BI/25/001/2006 BT/PR628/AGR/36/674/2011 DBT-BINC fellowship Council for Scientific and Industrial Research (CSIR), India fellowship This work was supported by the Department of Biotechnology (DBT, India) grant for funding the research and Bioinformatics Infrastructure Facility (BIF) at ICGEB (BT/BI/25/001/2006 and BT/PR628/AGR/36/674/2011). SS received a DBT-BINC fellowship and AK received a Council for Scientific and Industrial Research (CSIR), India fellowship. The funders had no role in study design, data collection and analysis, decision to publish, or preparation of the manuscript.

==============================
In the past decade, the microRNAs (miRNAs) have emerged to be important regulators of gene expression across various species. Several studies have confirmed different types of post-transcriptional modifications at terminal ends of miRNAs. The reports indicate that miRNA modifications are conserved and functionally significant as it may affect miRNA stability and ability to bind mRNA targets, hence affecting target gene repression. Next Generation Sequencing (NGS) of the small RNA (sRNA) provides an efficient and reliable method to explore miRNA modifications. The need for dedicated software, especially for users with little knowledge of computers, to determine and analyze miRNA modifications in sRNA NGS data, motivated us to develop miRMOD. miRMOD is a user-friendly, Microsoft Windows and Graphical User Interface (GUI) based tool for identification and analysis of 5′ and 3′ miRNA modifications (non-templated nucleotide additions and trimming) in sRNA NGS data. In addition to identification of miRNA modifications, the tool also predicts and compares the targets of query and modified miRNAs. In order to compare binding affinities for the same target, miRMOD utilizes minimum free energies of the miRNA:target and modified-miRNA:target interactions. Comparisons of the binding energies may guide experimental exploration of miRNA post-transcriptional modifications. The tool is available as a stand-alone package to overcome large data transfer problems commonly faced in web-based high-throughput (HT) sequencing data analysis tools. miRMOD package is freely available at http://bioinfo.icgeb.res.in/miRMOD.

Introduction

miRNAs are a class of small non-coding RNAs (ncRNAs) of size 20–24 bases, acting as post-transcriptional regulators of gene expression (Bartel, 2009). miRNAs and transcription factors are important components of gene regulatory networks (Gu, Zhang & Wang, 2012). Thus, any molecular modification in the regulators of integrated networks produces several downstream effects (Gu & Xuan, 2013; Wang, Gu & Li, 2014). It is established in plants as well as animal cells that different post-transcriptional processes modify miRNAs after biogenesis. The modifications processes include trimming, addition or substitution of miRNA nucleotides (Ebhardt et al., 2009; Kim, Heo & Kim, 2010). Modified miRNA with terminal additions may be of two types, namely templated or non-templated depending on its alignment or non-alignment with the reference genome, respectively. The non-templated additions (NTA) are biologically relevant and physiologically regulated processes mediated by enzymes (Wyman et al., 2011). Although, the functional significance of such post-transcriptional modifications is yet to be extensively explored, few studies have concluded that the modifications play an important role in miRNA stability and affects miRNA-target binding and hence the efficiency of target repression. For example, in humans-the terminal adenylation of miR-122 enhances the miRNA stability while terminal uridylation of miR-26A reduces its ability to inhibit its target (Jones et al., 2009; Katoh et al., 2009). In general, adenylation increases miRNA stability, whereas uridylation promotes miRNA degradation (Baccarini et al., 2011; Ibrahim et al., 2010; Katoh et al., 2009; Lu, Sun & Chiang, 2009). However, in few exceptional cases, as in humans and drosophila, both the 3′ adenylation and uridylation promote miRNA degradation (Ameres et al., 2010). It is known that amongst the two miRNA terminal modification types, the 3′ end modifications are more common as compared to the 5′ end modifications (Ryan, Robles & Harris, 2010). The less prevalent 5′ end modifications are of great interest as the modifications can modify the seed sequence of its reference miRNA and may alter target binding or even change target mRNAs (Ameres & Zamore, 2013). The 5′ modifications are also reported to be a widely conserved phenomenon, though its functional significance is yet to be explored (Ryan, Robles & Harris, 2010; Wu et al., 2007). Most of the reported miRNA modification studies primarily focus on terminal adenylation and uridylation, however there are fewer reports on modifications with cytosine and guanine (Aravin & Tuschl, 2005; Burroughs et al., 2010; Morin et al., 2008). Previous studies suggest that post-transcriptional modification in miRNAs is a highly regulated phenomenon and few miRNAs are comparatively more frequently subjected to modification (Wyman et al., 2011). Therefore, local and global analysis of miRNA modifications can enhance our understanding of miRNA medicated cellular gene regulatory mechanisms. The availability of affordable high throughput (HT) small RNA (sRNA) sequencing techniques has made it possible to determine and analyze modified sRNA sequences on a global scale. However, analyzing such HT data is not only a computational challenge but also relatively more difficult for biologists with non-computational academic background. Moreover, the literature survey revealed that currently very few computational tools are available to specifically study different types of miRNA modifications in sRNA NGS datasets (see Table 1). For example, isomiRex (Sablok et al., 2013) determines only the miRNA variants that align with the precursor miRNA sequences, whereas tools like miRanalyzer (Hackenberg et al., 2009) and miRGator (Cho et al., 2012) searches only 3′ non-templated modifications. SeqBuster is limited to determination of 3′ mofications and 5′ trimming (Pantano, Estivill & Marti, 2010). CPSS (Zhang et al., 2012) and Chimira (Vitsios & Enright, 2015) servers processes datasets for limited number of species only. Another tool, isomiRID (De Oliveira, Christoff & Margis, 2013), reports modified miRNA list without distinguishing templated and non-templated modifications. IsomiRage (Muller, Marzi & Nicassio, 2014) is a tool for quantifying user defined miRNA modifications in a small RNA NGS. Similar to the existing tools discussed above, the databases like YM500v2 (Cheng et al., 2013) also performs isomiR searches; however, it is restricted to data related to human cancers. Apart from the above-mentioned limitations, the web-based tools and databases also have technical and computational limitations like restriction of size and type of input files. The output files generated by most of the tools for analysis of miRNA modifications are simple text or html files, which more often requires manual efforts to obtain relevant statistics. Moreover, none of the tools focuses on the effect of modifications on miRNA-target interactions- an obvious and expected translational effect of such modifications.

Table 1 Comparison of miRMOD with other existing tools.

	5′ end modification	3′ end modification	Graphical display	Target alteration analysis	Stand alone	
	Addition	Trimming	Addition	Trimming				
miRanalyzer	No	Yes	Yes	Yes	No	No	Noa	
miRGator	No	No	Yes	Yes	No	No	Nob	
CPSS	Yes	Yes	Yes	Yes	Yesc	No	Noa	
isomiRID	Yes	Yes	Yes	Yes	No	No	Yes	
SeqBuster	No	Yes	Yes	Yes	Yesc	No	Yes	
miRMOD	Yes	Yes	Yes	Yes	Yes	Yes	Yes	
Notes.

a Restricted to selected species only.

b File size limit <20 MB.

c Limited features only.

The limitations of the existing tools motivated us to develop miRMOD package, which not only identifies and analyzes miRNA modifications (non-templated additions as well as trimming) but also performs local and global analysis of changes associated with modifications. miRMOD also provides an optional feature to predict the effect of miRNA modifications on miRNA-target affinities, which is consequential for its biological function.

Materials and Methods

Input file generation

Figure 1 is a summary illustration of miRMOD pipeline. miRMOD tool essentially requires three input files- namely, a fasta file of miRNA sequences, a fasta file of sRNA reads and Bowtie generated alignment file (Langmead et al., 2009). Prior to miRMOD run, the sRNA input file for Bowtie and miRMOD is required to be reformatted with the ‘prepare_input’ program, packaged with miRMOD. The ‘prepare_input’ module simply converts input file (fasta/fastq/tab separated format) into a miRMOD accepted preprocessed fasta format (No normalization is performed). The preprocessed file (output from ‘prepare_input’) should be used for Bowtie alignment with its reference genome or pre-miRNAs, using user-defined parameters. miRMOD package is bundled with three sample files, including a fasta file containing mature miRNA sequences in the human genome (2578 sequences; source: miRBase version 20).

Figure 1 miRMOD workflow.

Design and algorithm

The miRMOD algorithm uses the following criteria in order to search nucleotide additions/trimming in NGS datasets. The first criterion is that the input miRNA sequences must perfectly align with the corresponding reference genome. The second criterion is that the miRNAs reads with mono or poly-nucleotide additions at the 5′ or 3′ end of a miRNA sequence must not align with the reference genome.

For each miRNA, a Z-score is calculated to measure its relative tendency to get modified under given miRMOD run parameters. The score is calculated using the following formula: Z-score=si−μSσS

where, si represents the score for ith miRNA and defined as the product of number of modifications and total number of modified reads observed for a given miRNA. μ(S) represents the mean value of all miRNA scores (S) and σ(S) represents standard deviation of S.

Putative target sequences may also be specified as an optional input in order to compare target-binding affinities of query miRNA sequences with that of modified miRNAs. Target prediction is performed by RNAhybrid (Kruger & Rehmsmeier, 2006) using default parameters, followed by calculation of energy change and alteration in the target sites due to the modifications. miRMOD results are organized in different windows and tabs.

Sample datasets analysis and evaluation

In order to evaluate and check the robustness of miRMOD, we downloaded clean NGS reads of 15 libraries from a GEO entry with accession number GSE21279 (SRA: SRP002272). The reads of size 19–25 nucleotides and abundance equal or greater than 2 were selected for miRMOD analysis. The fasta formatted clean reads were converted into a miRMOD readable format, using miRMOD ‘prepare_input’. The ‘prepare_input’ processed reads were then mapped to the reference human genome using Bowtie (Langmead et al., 2009) . Only known miRNA sequences (2578 miRNA sequences, miRBASE v20) from each dataset libraries are used as the query sequences in order to identify miRNA modifications. The miRMOD output of miRNA modifications for all the 15 libraries was compared and validated with the modifications previously reported for the test case (Li et al., 2012).

Results

miRMOD facilitates identification and analysis of miRNA modifications in sRNA HT data. miRMOD identifies single and multiple additions/deletions of terminal nucleotides in miRNA reads. miRMOD analysis of already reported datasets illustrates the utility of the tool and helped evaluation of the package. Tabular display of output produced by miRMOD helps in visualization and analysis of miRNA modifications at global as well as local level. For each library, the first tab in the main miRMOD output summarizes the results in 3 sub-sections. The first sub-section displays the gross summary results of modified miRNAs in the dataset with SRA ID: SRX018958 (252 reads in this dataset), with their total number of modifications (1,699 reads) and the miRNA in the input dataset with the highest number of modified reads (Fig. 2). The length distribution graph shows that mono-nucleotide modifications are much higher than polynucleotide modifications. The second section of the tab gives the distribution of modified reads in four different types of modifications (3′ addition: 26, 75,808, 5′ addition: 8,512, 3′ trimming: 19, 81,380 and 5′ trimming: 70,237) and a graphical display of the results. The section also displays the percentage of modified reads present in input dataset (in this case it is 40%). The third section of the first tab summarizes the abundance of different type of sRNA reads i.e., miRNAs and the corresponding modified reads present in the input library, facilitating a global analysis. The ‘Details’ tab (Fig. 3B) gives comprehensive information about different types of modifications found for each query miRNA along with its modified form. The ‘Frequency table’ tab provides the percentage occurrence of each type of mono or poly-nucleotide modifications in the input dataset (Fig. 3B). A more detailed description about the utility and unique capabilities of miRMOD are discussed later with appropriate examples and comparisons.

Figure 2 miRMOD main output page for the sample library SRX108958.

Figure 3 miRMOD output from the additional tabs.

miRMOD accepts fasta sequences of putative or already established miRNA targets (e.g., 3′ UTR) to predict and compare minimum free energy and target site changes in miRNA:target and modified-miRNA:target interactions. This kind of analysis facilitates the prediction of modifications leading to target alteration, or strength of possible binding (Ryan, Robles & Harris, 2010).

miRMOD performance

We compared the miRMOD analysis of 3′ nucleotide additions with the results reported by Li and coworkers (Li et al., 2012), see ‘Sample dataset analysis and evaluation’ for details of miRMOD run. Notably, using miRMOD we were able to identify all 3′ additions (only non-templated) reported by them in each of the 15 libraries. Additionally, miRMOD also determines several other terminal modifications i.e., 3′ trimming, 5′ additions and 5′ trimming, not reported by them. Thus, miRMOD can help in identification and analysis of different types of novel modifications. We calculated the percentage of single (A, T, G and C) and di-nucleotide (A∗,T∗,G∗,C∗), 3′ and 5′ additions as well as trimming in each of the sample libraries (Tables 2 and 3). The percentage addition of ‘A’ at the 3′ miRNA end is the highest among all the other nucleotide additions in the libraries, followed by the ‘AT’ and ‘T’. Apart from ‘A’ and ‘T’ additions, miRMOD assisted analysis also revealed significant percentages of ‘C’ and ‘G’ nucleotide additions or miRNA trimmings, in addition to what is already reported in the publication related to the datasets (Li et al., 2012). The frequency of miR122-5p 3′ addition with adenylation is comparatively higher than that of any other miRNA in the libraries, in agreement with the previous reports (Li et al., 2012). As expected, the proportion of ‘C’ and ‘G’ addition at 3′ end is significantly less than ‘A’ or ‘T’ additions. However, the ‘C’ and ‘G’ nucleotides are trimmed mainly at 3′ end except in the sample SRX018961, where ‘A’ dominates all the other mono-nucleotide trimmings. Additionally, we also analyzed the nucleotide additions and trimming at the 5′ ends of miRNA sequences in all the sample libraries. It may be noted that there are comparatively few reports of 5′ modifications in the literature. The miRMOD analysis of the 15 libraries analyzed by us reveal that ‘C’ is the most frequently added nucleotide at the miRNA 5′ ends, followed by ‘A’ and ‘T’, in the order of decreasing frequencies. In the case of 5′ trimming, mono-nucleotide deletion of ‘C’ followed by ‘T’ surpasses all other nucleotide deletions. Thus, we found that ‘C’ is an important nucleotide involved in the 5′ end miRNA modification. The miRMOD analysis reveals that the most common terminal miRNA modification is 3′ additions, followed by 3′ trimming, 5′ trimming and 5′ additions, in the descending order of frequencies. As reported in previous studies, 3′ additions also include AA, TT and TA in significant proportion, which is also confirmed by miRMOD analysis of the dataset (Burroughs et al., 2010; Li et al., 2012). Apart from mono or di nucleotide modifications, miRMOD also determines polynucleotide additions and deletions at the miRNA terminal ends (Table S1); however, these form a very small fraction of the overall modifications.

Table 2 Frequency table of 3′ and 5′ additions.

(A) Frequency table of 3′ addition	
Library	A	AT	T	AA	TT	CA	G	TA	C	GT	GA	AG	GG	Others	
SRX018957	69.84	9.05	3.92	4.76	1.32	0.41	0.62	0.75	0.32	0.19	0.13	0.22	0.02	8.46	
SRX018958	74.39	7.8	3.72	3.05	1.25	0.32	0.6	0.55	0.18	0.17	0.08	0.1	0.01	7.8	
SRX018959	81.1	5.76	2.93	2.83	1.08	0.24	0.6	0.45	0.17	0.13	0.08	0.09	0	4.53	
SRX018960	76.98	5.68	5.26	4.27	0.79	0.55	0.58	0.37	0.35	0.2	0.11	0.18	0.01	4.65	
SRX018961	65.5	4.94	12.43	4.67	1.39	2.69	0.84	0.59	0.4	0.35	0.3	0.29	0.06	5.54	
SRX018962	72.49	8.35	6.08	3.65	1.02	0.42	0.69	0.56	1.27	0.18	0.17	0.16	0.12	4.82	
SRX018963	70.66	9.98	5.88	3.55	1.06	0.46	0.77	0.59	1.07	0.17	0.11	0.14	0.06	5.5	
SRX018964	64.92	11.2	6.92	4.03	1.55	0.62	0.76	0.73	1	0.2	0.14	0.17	0.13	7.64	
SRX018965	70.08	9.45	4.97	4.27	0.8	0.93	1.04	0.56	1.23	0.21	0.4	0.3	1.46	4.3	
SRX018966	77.6	6.05	3.66	3.35	1.34	0.5	0.53	0.62	0.14	0.16	0.08	0.1	0.01	5.85	
SRX018967	80.8	5.8	2.74	3.61	0.58	0.37	0.81	0.49	0.13	0.12	0.11	0.15	0.01	4.28	
SRX018968	81.68	4.1	3.46	3.64	1.01	0.6	0.52	0.57	0.25	0.16	0.21	0.14	0.02	3.62	
SRX018969	75.93	7.02	4.63	2.48	1.34	0.49	0.62	0.48	0.18	0.19	0.08	0.07	0.01	6.5	
SRX018970	76.57	5.62	2.67	2.65	1.84	0.22	0.46	0.47	0.16	0.15	0.07	0.09	0.01	9.01	
SRX018971	54.78	9.49	5.87	9.64	1.86	3.46	1.11	1.7	0.48	0.9	0.39	0.23	0.01	10.08	
Average	72.89	7.35	5.01	4.03	1.22	0.82	0.7	0.63	0.49	0.23	0.16	0.16	0.13	6.17	
(B) Frequency table of 5′ addition	
Library	C	A	T	TC	CA	CG	AA	AT	TG	G	Others	
SRX018957	76.57	5.76	6.52	5.84	0.93	0.78	0.43	0.23	0.13	0.05	2.77	
SRX018958	83.99	6.22	2.4	3.42	0.23	1.52	0.16	0.4	0.05	0.02	1.58	
SRX018959	86.74	5.38	2.28	3.31	0	0.78	0.12	0.23	0.09	0.03	1.05	
SRX018960	73.25	6.86	5.84	3.18	2.86	2.24	0.5	0.37	0.35	0.07	4.47	
SRX018961	70.99	8.16	8.08	1.6	1.86	2.13	0.9	0	0.4	0.32	5.56	
SRX018962	75.74	7.83	4.45	4.93	2.77	1.01	0.2	0.07	0.13	0.17	2.7	
SRX018963	78.71	7.41	3.27	5.01	1.85	1.09	0.19	0	0.14	0.14	2.18	
SRX018964	74.88	8	3.87	4.73	4.21	0.97	0.37	0.12	0.17	0.17	2.5	
SRX018965	72.75	9.33	3.22	4.78	5.9	0.73	0.3	0.14	0.05	0.21	2.61	
SRX018966	82.58	7.76	2.83	2.83	0.11	1.21	0.39	0.07	0	0.07	2.15	
SRX018967	77.91	11.29	2.99	2.39	0.19	0.45	0.62	0.29	0.04	0.04	3.78	
SRX018968	81.45	8.38	2.5	3.07	0	1.28	0.45	0	0.26	0.13	2.5	
SRX018969	74.06	13.86	3.8	2.79	0.58	1.3	0.77	0.24	0.14	0.14	2.31	
SRX018970	81.01	8.6	3.17	1.85	0	1.15	0.84	0.36	0.14	0.07	2.81	
SRX018971	85	6.93	1.4	2.13	0	0.36	0.9	0.12	0.08	0.08	2.99	
Average	78.38	8.12	3.78	3.46	1.43	1.13	0.48	0.18	0.14	0.11	2.8	

Another interesting observation is that miRMOD determined very few miRNAs (around 1–5) with terminal trimming that transforms it to another previously classified miRNA. For example, miR-548k is converted to miR-548av-5p in the sample SRX108971, due to removal of TCGT at the 3′ end. Although the percentage of such miRNA conversion is very low in the studied dataset, the studies on miRNA transformation in other sequencing datasets and experiments could be interesting.

We also observed a repetitive nature of modifications through the analysis of the 15 datasets. 1012 modifications are common amongst the 15 datasets. Few modified sequences in the datasets are present in higher abundance for example ‘1056282’ for has-miR-122-5p. Such a large number of repetitions of modified sequences rejects the probability that the sequences are mere artifacts and also suggests that such conserved modifications may be functionally significant and call for experimental scrutiny.

Comparison with other existing tools

Compared with the other tools, miRMOD is a user-friendly and easy to install, moreover miRMOD installation requires least software dependencies (see miRMOD manual). Unlike the currently available webservers and tools for miRNA modification analysis, miRMOD has the advantage that large dataset transfer over the Internet is not required and the analysis pipeline is species independent. The tool determines trimming and non-templated nucleotide additions at both terminal ends of miRNAs in sRNA NGS datasets. It also determines the percentage occurrence of each modification in input dataset, per miRNA modification score along with ‘miRNA to miRNA’ conversion.

The utility of miRMOD is best illustrated by a comparison of miRMOD analysis with other existing tools (Table 1). Comparison with CPSS tool requires detailed comparison, as it determines 5′ as well as 3′ trimming apart from addition of non-templated nucleotides. Amongst the other existing tools, miRGator requires input files in BAM format, with file size restriction of 20 MB. However, SeqBuster is a standalone tool which has several software dependencies required for its installation. miRMOD is free from any of the mentioned limitations and equipped with several unique features to facilitate and comprehensive analysis of miRNA modification in user defined inputs.

For comparison with CPSS, we used the sample dataset of human fetal ovary, available from CPSS website. Using miRMOD, we found that 355 miRNAs were modified from 668 miRNAs present in the dataset (Fig. 3A). The different types of modifications (5′ and 3′ non-templated additions and trimming with abundance ≥10, miRMOD default read count threshold) present in the CPSS output file were identified by miRMOD too. In the test library, hsa-let-7a-5p has the highest number of miRNA modifications with 485556 modified reads. Apart from this, the 3′ trimming (72%) is the most common modification type for the given sample dataset. The detailed result section further elaborated the summary for local analysis by separating the reads with different type of modifications (5′ or 3′) in a “TreeView” format as shown in Fig. 3B. This enabled us to infer the most prominent modifications for a selected modified miRNA without re-analyzing the predicted modifications as compared to the CPSS result. In the next section of miRMOD analysis, frequency table helped us in comparing the percentage occurrence of different modifications in the test dataset (Fig. 3C). This information can help us in identifying the most common nucleotides involved in modifications in the dataset and is also a unique miRMOD feature, not available in other tools. For example ‘GTT’ is a preferred nucleotides modification for 3′ trimming whereas ‘A’ is the most preferred modification for 3′ addition over the other types of modifications in the sample dataset. Another unique miRMOD feature ‘miRNA2miRNA conversion’ helps in identification of miRNA(s) that are trimmed and transformed into previously classified miRNA(s) from the list of miRNA(s) submitted by the user. miRMOD performs miRNA scoring and prioritization based on its tendency to modify and produce modified reads (Fig. 3D, see ‘Materials and Methods’), a unique feature of miRMOD tool. Thus, the information like which miRNAs has the higher tendency for modification as compared to other miRNAs is swiftly retrievable from the dataset using miRMOD, unlike other prediction tools. Moreover, the optional and novel miRMOD feature of Target Variation Analysis (TVA) allows multi-way capturing of highly modified miRNAs and its modification(s) to explore if selected modification is playing any role in altering the miRNA-target interactions (Fig. 3E). Thus, miRMOD offers several unique features for sRNA data analysis, which makes it highly useful tool for modification analysis.

miRMOD evaluation with experimentally validated miRNA modifications

We used miRMOD to identify experimentally validated miRNA modifications from a publicly available dataset of human neural dataset (SRA: SRX548700) (Tan et al., 2014). Not surprisingly, miRMOD successfully identified hsa-miR-9-1 miRNA sequence as well as its experimentally verified modified miRNA ‘CTTTGGTTATCTAGCTGTATGA’ (generated as a result of 5′ trimming) in the sRNA dataset. Apart from the above-mentioned modification, two additional modified sequences were also reported by miRMOD for hsa-miR-9-1 viz. 3′ trimming and 3′ addition of ‘A’ nucleotide (see Supplemental Information 1). The execution time for complete analysis of a given miRNA using the benchmark dataset was ∼15 min.

Table 3 Frequency table of 3′ and 5′ trimmings.

(A) Frequency table of 3′ trimming	
Library	G	TG	A	C	T	GA	TT	GT	CA	CC	GC	TA	AA	CT	AG	CG	Others	
SRX018957	38	14.4	5.69	11.4	13.3	0.49	3.17	0.26	0.03	0.33	0.2	0.45	0.37	0.24	0.16	0.04	11.5	
SRX018958	37.1	7.38	12.2	12	7.27	2.55	1.76	1.08	0.76	0.9	0.21	0.35	0.48	0.22	0.14	0.12	15.56	
SRX018959	45.6	7.87	8.13	10.3	6.64	2.48	1.63	1.05	0.82	0.74	0.2	0.24	0.36	0.16	0.14	0.08	13.61	
SRX018960	31.6	14.9	19.6	10.6	5.67	0.64	1.41	0.76	0.7	0.24	0.43	0.79	0.58	0.22	0.2	0.01	11.62	
SRX018961	20.3	23	22.3	8.27	6	1.65	1.95	0.79	0.58	0.28	0.45	0.92	0.33	0.25	0.18	0.03	12.8	
SRX018962	33.3	17.8	9.58	7.49	5.51	1.39	2.06	2.25	1.04	0.35	0.63	0.37	0.4	0.2	0.23	0.03	17.35	
SRX018963	30.6	15.4	12.8	7.65	8.75	1.23	2.41	1.78	1.02	0.38	0.54	0.54	0.39	0.22	0.21	0.05	16.12	
SRX018964	33	13.2	12.3	8.85	8.82	1.34	2.12	1.55	1.02	0.37	0.63	0.42	0.5	0.28	0.18	0.05	15.36	
SRX018965	34.4	12.8	8.28	8.65	3.9	2.8	0.48	5.93	1.93	0.47	0.54	0.15	0.31	0.9	0.32	0.31	17.87	
SRX018966	42.6	8.14	10.7	8.5	8.49	4.06	1.98	1.11	0.75	0.74	0.25	0.28	0.3	0.27	0.11	0.16	11.62	
SRX018967	50.5	7.96	4.69	13.8	4.57	4.42	1.08	0.76	0.81	0.94	0.43	0.1	0.1	0.13	0.07	0.08	9.59	
SRX018968	40.8	11.2	7.52	5.14	8.62	4.04	3	1.66	0.79	0.52	0.22	0.31	0.29	0.37	0.19	0.21	15.21	
SRX018969	45.8	13.2	7.02	8.51	4.66	3.89	1.34	1.32	0.58	0.72	0.21	0.24	0.27	0.19	0.11	0.12	11.82	
SRX018970	41.9	7.62	12.8	9.04	6.1	3.59	1.4	1.23	0.81	0.71	0.22	0.39	0.51	0.3	0.15	0.21	13.07	
SRX018971	15	3.36	11	11.5	12.9	13.9	2.9	2.26	2.29	1.02	0.45	0.06	0.21	0.67	1.34	0.35	20.81	
Average	36	11.9	11	9.43	7.41	3.23	1.91	1.59	0.93	0.58	0.38	0.37	0.36	0.31	0.25	0.12	14.26	
(B) Frequency table of 5′ trimming	
Library	C	T	A	TG	TC	CT	TA	AC	G	Others	
SRX018957	46.75	22.67	23.95	3.15	0.89	0.98	0.3	0.36	0.07	0.87	
SRX018958	25.51	47.59	21.75	0.67	1.77	0.62	0.71	0.29	0.09	1	
SRX018959	30.2	48.28	17.47	0.44	1.28	0.64	0.51	0.24	0.09	0.86	
SRX018960	46.4	17.25	30.41	2.2	1.33	0.85	0.41	0.57	0.07	0.53	
SRX018961	45.27	14.12	36.35	1.05	0.78	0.43	0.46	0.71	0.08	0.76	
SRX018962	44.01	27.73	16.39	7.04	1.35	1.4	0.47	0.3	0.08	1.24	
SRX018963	44.23	23.33	22.61	5.12	1.13	1.5	0.36	0.5	0.06	1.14	
SRX018964	50.41	21.29	21.43	3.03	0.99	0.86	0.33	0.41	0.06	1.18	
SRX018965	47.64	36.72	4.33	6.09	1.71	0.94	0.46	0.07	0.18	1.86	
SRX018966	39.39	35.58	20.64	0.56	0.97	0.49	0.59	0.24	0.1	1.44	
SRX018967	62.41	30.08	4.44	0.28	0.85	0.87	0.34	0.04	0.12	0.56	
SRX018968	61.82	19.63	15.27	0.52	0.77	0.34	0.39	0.14	0.15	0.98	
SRX018969	71.2	21.41	4.36	0.43	0.54	0.73	0.38	0.06	0.2	0.68	
SRX018970	32.39	47.83	15.15	0.67	1.45	0.35	0.81	0.13	0.12	1.1	
SRX018971	61.53	19.77	13.64	0.4	1.57	0.49	0.55	0.03	0.47	1.56	
Average	47.28	28.89	17.88	2.11	1.16	0.77	0.47	0.27	0.13	1.05	

Conclusions

miRMOD is a Windows-based standalone specialized tool to identify and analyze non-templated nucleotide additions and trimming at both the termini of the miRNA sequences in sRNA NGS data. The tool provides useful statistics about various types of miRNA modifications along with its frequencies, differential miRNA-modified miRNA abundance and frequencies of the modified bases found in each of the modifications. miRMOD also analyzes differences in miRNA target interactions, as a result if miRNA modifications. The miRMOD analysis gives a global as well as micro overview of miRNA modifications in sRNA NGS datasets. miRMOD is available as a GUI based as well as cross-platform command-line package, to generate user-friendly graphical representations for miRNA modifications in a NGS sRNA dataset, and its effect on miRNA-target binding affinity.

Availability and Requirements

Project name: miRMOD

Project home page: http://bioinfo.icgeb.res.in/miRMOD

Operating Systems: Windows XP, Windows 7, Windows 8 (recommended)

Programming language: C#

Other requirements: .NET framework 4 and RNAhybrid (optional)

License: Free

Supplemental Information

Table S1 Frequency table of all types of modifications in SRA dataset: SRP002272

Click here for additional data file.

Supplemental Information 1 miRMOD sample session file

Click here for additional data file.

Additional Information and Declarations

Competing Interests

Author Contributions

Data Availability

The authors declare there are no competing interests.

Abhinav Kaushik and Shradha Saraf performed the experiments, analyzed the data, contributed reagents/materials/analysis tools, wrote the paper, prepared figures and/or tables, performed the computation work, reviewed drafts of the paper.

Sunil K. Mukherjee conceived and designed the experiments.

Dinesh Gupta conceived and designed the experiments, analyzed the data, contributed reagents/materials/analysis tools, wrote the paper, performed the computation work.

The following information was supplied regarding data availability:

The source code and installation instructions are available at http://bioinfo.icgeb.res.in/miRMOD.

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
