# Peer review of "miRMOD: a tool for identification and analysis of 5′ and 3′ miRNA modifications in Next Generation Sequencing small RNA data"

_PeerJ, doi:10.7717/peerj.1332_

## Round 0.1 · original submission · Major Revisions

The manuscript has been assessed by two independent external reviewers who found merit in your work but also pointed out a list of issues that need to be properly addressed before advising for publication.

Reviewer 1 ·

Basic reporting

na

Experimental design

na

Validity of the findings

na

Additional comments

Kaushi and co-workers present an interesting manuscript entitled “miRMOD: A tool for identification and analysis of 5′ and 3′ miRNA modifications in Next Generation Sequencing small RNA data”. In their work, authors address an important topic, the detection of 3’ and 5’ modifications in small non-coding RNAs such as miRNAs. While I am generally convinced that such user-friendly tools can support our efforts to discover novel aberrations in miRNAs that influence target binding and may have pathological influences, the current work is not acceptable in the present form and need substantial revision. Among the most important points are:
1) Authors should have a comprehensive list of other tools that do the same job. Just claiming that “few computational tools with limited capabilities” exist is by far not enough, without a complete list and a comparison such a statement is just devaluing the work of others and does not contribute. Some tools that can predict modifications in miRNAs include IsoMirage, miFram or YM500 / YM500v2.
2) The second concern is the direct comparison. The current section in the manuscript does not describe a tool-by-tool comparison. This may also not be necessary, but at least for one tool authors should present a very detailed comparison. I would highly recommend mirage, one of the best tools in our hands. Just by describing a clear add-on such strong statements ad in remark 1 are justified.
3) The third concern is the section starting with “in order to validate”. Here, readers expect to actually get a description of a validation. Running available data sets through a pipeline is by no means what non-bioinformatic readers may understand using the term validation. Indeed, authors should not only change “validation” to a more adequate term such as “evaluation”, moreover, authors should indeed validate the algorithm. If miRMOD is superior to other algorithms, few selected novel modifications should be repeated in an experiment. I fully understand that a complex validation that proofs that binding affinity to targets is disturbed is beyond the scope of such a manuscript, however, demonstrating in a low-throughput manner that the modifications are no artifacts by NGS is feasible and from my perspective a pre-requisite.
4) In general, authors should correct and proof-read their manuscript carefully. Claiming e.g. that “recent studies” suggest something and then citing a manuscript from 2011 is misleading. This is however just one random example of respective small imprecisions.

Again, I want to point out that I am a big supporter of respective computer aided tools. I think the presented approach has some nice features and would be happy to evaluate a revised manuscript by Kaushi and co-workers.

Reviewer 2 ·

Basic reporting

No Comments

Experimental design

No Comments

Validity of the findings

No Comments

Additional comments

The manuscript describes a bioinformatics tool, miRMOD, to identify miRNA modifications from small RNA next generation sequencing data. The tool is implemented in Windows OS with GUI. It is easy to install and run, which is particularly useful for non-bioinformatics audience. The output summary or reports are very useful with appealing looks. The drawback is the low scalability of the Windows application and the limit of per sample based analysis, which is not very practical for a larger number of samples. Some specific questions are below:
1) Small RNA-seq can be noisy and artificial modifications, either from sequencing or alignment, can occur. How do the authors make sure the identification of real modifications, not the false ones, particularly for the single nucleotide change, the biggest category in the findings?
2) Most of the modifications can be just isoforms of miRNAs and may not have functional impact. It would be important to highlight the modifications that indeed potentially affect miRNA functions.
3) Target prediction and free energy change calculation are described but the function of the application appears not active or at least from the output of my testing?
4) The authors should consider providing the command line option of the application to reach the broader users.
5) Once the graphic result display is closed, how the user can bring it back? It appears that function is lacking.
6) There are some grammar and sentence errors in the manuscript that needs to be carefully proofread.

---

## Round 0.2 · accepted · Accept

The concerns raised by the reviewers have been adequately addressed and the manuscript can be now accepted for publication.

Reviewer 2 ·

Basic reporting

No comments

Experimental design

No comments

Validity of the findings

No comments

Additional comments

The authors adequately addressed the concerns/issues in this revision. I have no further comments.